# Consolidating Port Decarbonisation Implementation: Concept, Pathways, Barriers, Solutions, and Opportunities

**Anas S. Alamoush *** , **Dimitrios Dalaklis** , **Fabio Ballini and Aykut I. Ölcer**

Maritime Energy Management Specialization, World Maritime University, SE 201 24 Malmö, Sweden; dd@wmu.se (D.D.); fb@wmu.se (F.B.); aio@wmu.se (A.I.Ö.)
* Correspondence: asa@wmu.se

**Abstract:** Industries worldwide are facing the urgent need to decarbonise in alignment with the goal of the Paris Agreement (PA), which aims to limit global warming. However, progress towards achieving this extremely important goal has been sluggish, and the wider maritime transport sector (ports included) is no exception. Despite practical barriers faced by ports, solutions have not yet been developed. Similarly, the definition or concept of decarbonisation, including opportunities arising for the port sector, remains underdeveloped and lacks support from academic research. Specifically, there are a lack of conceptual studies that yield clear and usable results and evidence. To address this gap and shed light on port decarbonisation as a contemporary issue (the study aim), this study has conducted a literature review consulting different academic and grey studies. The results of this study define the concept of port decarbonisation, highlight the barriers that hinder progress in this area, and establish solutions to guide ports in implementing decarbonisation measures and mitigating barriers. Building upon these findings, this study not only contributes to scholarly discussions surrounding port decarbonisation but also offers valuable implications for port managers, policy makers, practitioners, and other pertinent authorities. By properly understanding the concept of decarbonisation and its barriers and expanding knowledge in relation to it and its practical implementation, including the eye-opening opportunities, port stakeholders can actively contribute to the objective of the Paris Agreement and the broader pursuit of sustainability.

**Keywords:** decarbonisation; ports; ships; barriers; definitions; implementation; measures; solutions

## 1. Introduction

Ports play a crucial role in cities' and urban areas' economic, environmental, and social sustainability, with the impact they have extending well beyond the waterfront. They are vital nodes in the urban logistics network, enabling the movement of products and commodities into and out of towns and cities. Ports considerably boost urban economies by creating employment opportunities, bolstering trade-related industries, and increasing local tax revenues. According to a plethora of research that highlighted ports' role in cities and urban areas, including energy transition and decarbonisation, e.g., Refs. [1–4], ports play a vital role in freight transportation, serving as crucial transfer and consolidation nodes that optimize supply chains and reduce transportation costs for urban businesses. Additionally, the strategic location and role of ports in integrating global supply chains makes them indispensable elements of urban logistics, ensuring that cities remain economically robust and well-connected to the international markets.

Ports are entering a new stage that requires them to expand beyond their typical role in cargo handling and value-added logistics. This is attributed to pressing environmental concerns, particularly the decarbonisation and energy transition of the transport sector, including the associated strict regulations and scrutiny. Maritime transport is dependent on fossil fuels [5]. Thus, its decarbonisation, in line with Paris Agreement goal to limit global warming, is at the top of countries' and industries' agenda, considering that not meeting

the decarbonisation goal intensifies the climate change threats to the environment and humanity [6]. This is manifested by provisions, directives, and guidelines from intergovernmental and non-governmental organisations, e.g., International Maritime Organisation (IMO) [7–10], International Association of Ports and Harbours (IAPH), and the World Port Sustainability Programme (WPSP) [11–13], as well as the European Commission's Climate Law, the European Seaport Organisation (ESPO) [14], and the Association for Waterborne Transport Infrastructure (PIANC) [15].

Today, ports are urged to decarbonise more than ever due to several fundamental drivers and motivations. Decarbonisation meets the global and regional regulations (compliance) [16], e.g., Paris Agreement, European Union (EU) green deal, and climate law, including the Onshore Power Supply (OPS), alternative fuels bunkering (Directive 2014/94/EU), and Emission Trading Scheme (EU ETS). Similarly, taking environmental actions reduces the pressure from surrounding communities, NGOs, logistics chains (customers, shippers, consignees, liners, and carriers), insurers, politicians, and city officials [17–19]. It also greens the port and expands its sustainable performance [20,21], which ultimately improve economic competitiveness [22]. Furthermore, it is argued that decarbonisation contributes to meeting the ports' corporate social responsibility (CSR) [23,24], improves energy efficiency and security (through renewable energy and energy efficiency) [25], and decreases energy costs [26]. These in turn fortify the ports' green image and attract young and talented professionals who meet the requirements of the technological revolution. Also, port decarbonisation contributes to achievement of the environmental dimensions of the United Nations Sustainable Development Goals (UN SDGs, 2030 agenda), i.e., Goal 7 (access to renewable energy), Goal 12 (sustainable consumption and production), and Goal 13 (actions to mitigate climate change), among others [27]. Similarly, port decarbonisation strengthens the commitment to IMO's GHG strategy for shipping decarbonisation [10,28].

Several studies have addressed port decarbonisation (GHG emissions), though in a fragmented way and from different perspectives, i.e., technical [29], operational [30], management and policies [31], energy efficiency [32], ship side operations [19,26]), and land transport [33]. Additionally, numerous reviews have focused on port decarbonisation and energy efficiency, resulting in a collection of various measures and solution including policies [28,34–38].

The problem, from a practice and technical perspective, is that, despite the visible impact of climate change and pressing regulations, industries, including shipping [39,40], ports [34,36,38] and land transport [41], have not yet achieved their target to decarbonise and reach zero emissions. Maritime transport is thus moving at a snail's pace in the decarbonisation process, although its share in global GHG emissions is high, i.e., ships emit around 3% [40] and ports around 2% [5] of global $CO_2$ emissions. This slow uptake of decarbonisation technologies and measures, exacerbated by the expected increase in freight transport and its emissions [40,42], is attributed to various barriers that restrict adoption of technologies, i.e., organizational, institutional, economic, political, regulatory, managerial, and technical barriers, among others [37,43,44]. Academically, a broad overview of port decarbonisation studies indicates that barriers were referred to generically and no specific study investigated the barriers from a port perspective, in particular decarbonisation and energy transition. No study has defined port decarbonisation and linked it to the Paris Agreement. Another academic gap is that, despite the barriers problem, no study thus far has provided sound solutions and suggestions to break down and mitigate their effects. Based on the identified problems and gaps, while considering the environmental pressure on ports, this study, via a thorough literature review, aims to define the port decarbonisation concept and pathways and provide unified categorisation of the barriers and solutions to port decarbonisation while identifying opportunities arising for ports from riding the track of decarbonisation.

The novelty of this study is that it is the first study to shed light on the port decarbonisation concept (specifically) and to identify barriers and solutions, which contribute to filling academic gaps (immature area) by enriching the literature with categorisation and solutions that can be used in academic cross-pollination. Importantly, the study inspires port practitioners and policy and decision makers about the barriers and gives them answers (solutions) for their concerns while, at the same time, suggesting areas to focus on and what they need to reconfigure and improve. This also contributes to the Paris Agreement and port sustainability implementation.

The outline of the article is as follows. While Section 1 presented the research problem, gaps, the study goal, methods, and brief implications, Section 2 presents materials and methods, and Section 3 presents the port decarbonisation concept, definitions, and pathways. Section 4 presents the categorisation of the barriers while Section 5 proposes solutions to mitigate the barriers. Section 6 presents the opportunities for ports, and the final Section 7 contains the discussions and conclusions.

## 2. Materials and Methods

This study conducted a comprehensive literature review by searching Scopus, Web of Science, and Google Scholar databases. Combinations of search terms were used iteratively, i.e., (port OR sea port OR terminal) AND (decarbonisation OR $CO_2$ OR GHG OR greenhouse gas OR carbon OR emissions) AND (reduction OR strategy OR measure OR plan OR Barriers OR challenge). This yielded 241 studies. Duplicate studies were removed (99). The filtering process included two stages. Stage one focused on exclusion of irrelevant studies based on title and abstract reading; thus, those that were generic in nature (33) and did not meet the review question (23) were removed. The second stage analysed the paper by full reading; hence, only high-quality studies were included, particularly those that met the objective of this study and had scientific rigor. Additionally, repetitive and low-quality studies were not considered. As a result, 86 studies were included. As regards the decarbonisation concept, the Paris Agreement and the IPCC reports were considered. With respect to the decarbonisation pathways, rigorous and high-quality systematic literature review studies were taken into consideration because categorisations were already established. For both the identification of the drivers and barriers, the guidelines in [45] for building typologies (taxonomies) were adopted. Building typologies requires researchers to identify critical dimensions of a concept to reconcile conflicting findings from previous research while at the same time organising fragmented research into common distinct types. While some barrier typologies were already established, several new categories were built and added in this study. The solutions and opportunities were built innovatively in an inductive way. As a result, this study identified eight barriers and thirteen solutions that, one way or another, mitigate these barriers. See Figure 1 for the typologies (taxonomies) of barriers and solutions. Overall, the conceptual classification and taxonomies lay the foundation for new theorising and also enable further empirical analysis.

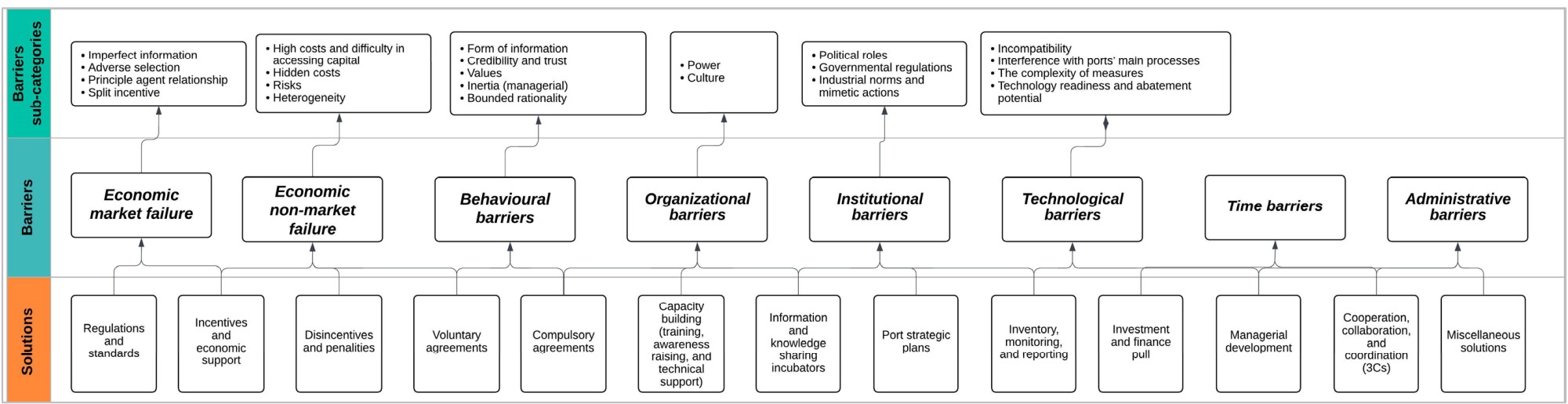

**Figure 1.** Typologies of barriers to port decarbonisation and their solutions.

### 3. Decarbonisation Concept and Definitions

The primary greenhouse gases (GHGs) that have impact on the climate are water vapor, carbon dioxide ($CO_2$), nitrous oxide ($N_2O$), methane ($CH_4$), and ozone ($O_3$). Their increase intensifies the greenhouse effect that warms the surface of the troposphere. When infrared radiation is emitted by the Earth's surface, GHGs absorb and re-emit them back to earth rather than permitting them to pass through into space. Thus, global warming occurs, which trigger climate hazards that cause severe impacts, e.g., warming, precipitation, floods, drought, heatwaves, fires, sea level rise, storms, diseases such as global cholera, and water supply contamination [46]. These hazards influence six key aspects of human life, i.e., health, food, water, infrastructure, economy, and security [46].

To stop the future climate change impacts, the Paris Agreement aims to limit global warming, preferably to 1.5 °C and significantly below 2 °C by 2100, compared to the pre-industrial level [47]. This requires global sectors to decarbonise on average by 2050 [48]. In the Paris Agreement, COP26, more than 140 countries, covering 90% of global GHG emissions, put forward the net zero goals (climate neutral) by 2050, with ambitious commitments by 2030 [47,49]. The commitment is to be achieved through short- and long-term sustainability measures, although there are slightly different timeframes and benchmarks between countries and sectors as they start from different baselines. Because all sectors need to decarbonise by 2050, the 2050 benchmarks are similar across all countries, whereas the 2030 benchmarks provide an interim step on the pathway towards 2050 [48]. This means that GHG emissions must be reduced considerably by 2030 (low and near zero), while achieving net zero emissions (climate neutral) is by 2040 or 2050 at latest.

In this sense, **decarbonisation** is the achievement of net zero GHG emissions by 2050. **Net zero** (In net zero, actors need to reduce their absolute GHG emissions across its whole supply chain (i.e., stakeholders in scope 3, such as suppliers, investments, employees), directly or indirectly, in order to support the target to limit global temperature increases to 1.5 °C), as underlined by the Paris Agreement, is to "achieve a balance between anthropogenic emissions by sources and removals by sinks of greenhouse gases in the second half of this century (2050)" [50]. In other words, to reach **net zero**, countries need to reduce GHG emissions as much as possible by mitigation measures, and the surplus emissions should be balanced by removals, such as removal of $CO_2$ emissions from the atmosphere by carbon sequestration processes. Decarbonisation has become a worldwide imperative and a top priority for governments, businesses, and society as a whole due to its vital role in reducing global warming. Mitigation measures include the switch from fossil fuels such as coal, natural gas, or oil to carbon free renewable energy technologies and energy sources such as low carbon fuels.

The Sixth Assessment Report of the Intergovernmental Panel on Climate Change (IPCC) provided warnings with regard to the increase of anthropogenic $CO_2$ emissions; the report called for urgent global actions to combat climate change [51]. According to the report, it is evident that other GHGs and air pollutants may impact the climate, but **carbon dioxide ($CO_2$)** remains the primary cause of global warming, and thus carbon emission is a worldwide issue in the twenty-first century and should be restrained and controlled. On this basis, the term used to achieve net zero carbon emissions is "carbon neutral". Derived from [52,53], **carbon neutrality** is achieved when a polluter's net contribution to global $CO_2$ emissions is zero, meaning that complete decarbonisation is reached, mainly with direct or indirect reduction of $CO_2$. In other words, the $CO_2$ emissions from the polluter's activities need to be fully compensated by reducing the $CO_2$ or removals entirely claimed by polluter, irrespective of the of the time period or the relative magnitude of emissions and removals involved. The term **carbon offset** is also used in climate policy which refers to a unit of $CO_2e$ (Carbon dioxide equivalent ($CO_2e$): A way to place emissions of various radiative forcing agents on a common footing by accounting for their effect on the climate. It describes, for a given mixture and amount GHGs, the amount of $CO_2$ that would have the same global warming ability, when measured over a specified time period". Thus the $CO_2e$ is the GHG emissions assuming a 100-year global warming potential [52])

emissions that is reduced, avoided, or sequestered to compensate for emissions occurring elsewhere. Additionally, "**carbon positive**" can be defined as "residual emissions will need to be offset or inset, i.e., cross-sectoral offsetting, or insetting in the same sector", while "**carbon negative**" can be defined as "net reduction in $CO_2$ through generation of surplus of renewable energy or carbon sequestration" [53].

Based on this section discussion and a comprehensive literature review, this line of research has built definitions for decarbonisation of maritime transport (shipping and ports), which are as follows:

- *Decarbonisation* is defined in this study as achievement of net zero $CO_2$ emission by 2050 by using mitigation measures and/or through the balance of surplus emissions by removal (e.g., carbon sinks and sequestration). "Mitigation measures" indicates the switch from fossil fuels such as coal, natural gas, or oil to carbon free renewable energy technologies and energy sources such as low carbon fuels.

- *Maritime transport decarbonisation* can be defined as the process of eliminating ships' $CO_2$ and other GHG emissions through mitigation measures or balance of surplus emissions by removal leading eventually to net zero $CO_2$ emission by 2050. The IMO, based on the recently adopted GHG strategy [10], pledged to reduce the total annual GHG emissions from international shipping by at least 20%, striving for 30%, by 2030, and 70%, striving for 80%, by 2040, respectively, compared to 2008; that is to say that the industry aims to reach net zero emissions by or around, i.e., close to, 2050.

- *Port Decarbonisation* is defined as the utilization of mitigation measures (technical and operational emission reduction measures) to reduce, neutralise, and offset $CO_2$ emissions from various port emission sources (port operation, ships, and land transport), while surplus $CO_2$ emissions are offset by sinks or sequestration; that is to say that the industry aims to reach net zero emissions by 2050 in line with Article 2 of Paris Agreement.

*Technical and Operational Measures for Port Decarbonisation*

Measures to decarbonise emission sources in the port (GHG emission reduction) have been reviewed and analysed, e.g., review of the tools for port sustainability [21], port energy efficiency measures [34], European ports' energy efficiency best practices [35], ports' technical and operational measures to reduce GHG emissions [36], port polices for decarbonisation [37], ports' measures for shipping decarbonisation [28], and solutions for planning the zero emission port through energy efficiency [38].

As can be seen in Table 1, there are various decarbonisation measures. The list of port decarbonisation measures is as per the categorisation in [36], which includes measures to decarbonise port operation (i.e., cargo handling equipment, e.g., cranes, derricks, forklifts, port trucks, and vehicles, etc.) and infrastructure and other marine service activities (e.g., tugs, tow and pilot boats). For in-depth details, readers are instructed to read the same study [36]. Additionally, there are measures ports can take up to reduce carbon emissions from land transport, as well as from ships at the ship-port interface.

**Table 1.** Port decarbonisation technical and operational measures.

| Scope | Categories | Measures |
|---|---|---|
| Port operation | Equipment | **Replacement, repowering, and refitting** of older carbon emission sources such as equipment or engines |
| | Energy sources | **Alternative fuels such** as LNG, methanol, hydrogen (fuel cells), ammonia, and biofuel |
| | | Development of biofuel **production facilities** to produce biogas, and biofuels |
| | | **Alternative electrical power systems** (either alternating current, or direct current through batteries), and hybridisation (battery and fuels) |
| | | **Renewable energy utilisation** (solar, wind, ocean, and geothermal energy) |

**Table 1.** *Cont.*

| Scope | Categories | Measures |
|---|---|---|
| Port operation | Energy efficiency | **Energy saving measures** for buildings, warehouses, storages, yards, harbour craft and marine services, reefers, cargo handling equipment, and employees' commuting |
| | | **Use of energy management systems and plans** e.g., ISO 50001, EN 16001, EN 16258 [54–56] |
| | | **Energy management technologies** (e.g., energy storage systems, reclamation, smart grids, virtual power plants, and microgrid) |
| | Operational measures | Container **terminal automation** |
| | | **Container terminal operation system** (TOS) |
| | | **Maintenance** of port equipment |
| | Port city integration and industry interactions | Harnessing the **circular economy** for waste management, recycling processes, and reuse of heat and steam (the ecology of scales) |
| | | Harnessing **Carbon Capture Storage** (**CCS**) and utilisation (CCU), carbon sinks |
| | Digitalisation | The use of digital transaction technologies (**digitalisation**) (e.g., Electronic Data Interchange (EDI), E-business, port community system, and single window) |
| | | Use of **smart and intelligent technologies**, (e.g., Internet of Things (IoT), block chains, 5G technologies) |
| Land Transport | Truck emission reduction | **Bans** on old trucks, use of intelligent transport systems, off-dock staging yards and shore power |
| | Modal shift/split | **Support moving** cargo by rail, barges, and inland waterways, **Motorways of the Seas** (MoS), **truck platooning**, and developing **dry ports**, and investment in **intermodal rail and barge facilities** and superstructure |
| | Truck congestion reduction | **Truck appointment** system (TAS) or vehicle booking system (VBS) |
| | | The use of **smart gates** |
| | | Extending **off-peak terminal** and gate hours |
| | | Imposing a peak hours **traffic mitigation fee** (TMF) |
| Ship-Port Interface | Ships' berth emission reduction | Provision of **Onshore Power Supply** (OPS) (electricity) at berth, including **hyper chargers** |
| | | Provision of electricity through LNG or fuel cell-based barge and/or **mobile generators** |
| | | Barge-based **emissions collection** |
| | Alternative fuels bunkering | Alternative (cleaner) fuels **bunkering** for ships, e.g., LNG and methanol, ammonia, and hydrogen, etc. |
| | Ship turnaround time reduction | Utilisation of **terminal operating systems** for berth allocation, yard equipment allocation and scheduling |
| | | Use of **automated mooring systems** (AMS) |
| | | Allocating **berth windows** by booking of berths before arrival |
| | | Utilisation of **mid-stream operations** (loading and unloading of cargo containers between ships at non-berth locations) |
| | Vessel Speed Reduction (VSR) | Vessels need to **reduce their speed** when arriving and departing the port |
| | Virtual and Just-In-Time Arrival | Managing the ships **Virtual and Just-In-Time Arrival** through utilisation of electronic data exchange, Port CDM, among other digital technologies |
| | Provision of Miscellaneous services | Provision of **ships' hull cleaning** and propeller polishing service |
| | | Provision of **electric shore-side pumps** for bulk liquids |

It should be noted that decarbonization measures are not a silver bullet to achieve zero emission ports. However, they can assist in the transition through combinations of measures in order to achieve the maximum abatement potential. In addition, the measures have challenges (technical issues, costs, and varying abatement potential, space requirement (high density of alternative fuels), high upfront cost, environmental benefits (the case of

electrification based on fossil fuels grids), life cycle issues, methane and $CO_2$ slip, the long lead time for infrastructure upgrading and renovations, etc.); see explanation of the barriers, in detail, in Section 4. Consequently, ports should carefully evaluate the measures and consider the advantages and disadvantages through feasibility and cost benefit ratio studies.

## 4. Barriers to Decarbonisation

Although ports can reach their climate targets due to availability of technologies (see Table 1), they are lagging behind due to different barriers that result in an implementation gap. An implementation gap, or paradox, appears once what is expected (objective of a policy) and the outcome are compared and the result leads to the observation that there is an implementation gap (failure) [57]. Sorrell et al. described the barriers as inhibitors that restrain investments in environmentally and economically efficient technologies and measures [58,59]. Barriers influence proper implementation despite existing stringent requirements (regulations). Most of the barriers, particularly to energy efficiency, have been investigated from the perspective of neoclassical economic theory, such as agency theory and contract theory, e.g., Refs. [58–60], though there were some investigations from the perspective of shipping energy efficiency [43,61,62], including decarbonisation [63]. Sorrell et al. built a framework to investigate the barriers, i.e., organisational (power and culture), behavioural (bounded rationality, form of information, credibility and trust, inertia, values and priority accorded), economic market failures (principal agent problem, split incentives, moral hazard, imperfect/asymmetric information), and non-market barriers (market heterogeneity, hidden cost, access to capital and risks) [58,59]. Thus, this study has adopted these categories to investigate the barriers to port decarbonisation, in addition to including some new categories identified in the review (see Figure 1).

In the following section, we provide a systematic analysis of barriers (Figure 1) from the perspective of port decarbonisation following the framework of Sorrell et al. [58,59] and augmented by research that has studied barriers from theory and practice, i.e., [60,64–74].

### 4.1. Economic Market Failure

**Imperfect information**

This barrier refers to the lack of right and proper information about decarbonisation technologies (e.g., abatement potential, fuel saving, cost benefit ratio). Consequently, while such asymmetric information increases uncertainty, it also leads to loss of opportunities to adopt cost-effective technologies owing to such distorted information. The information on decarbonisation measures may be subject to sellers' opportunism: they may provide right information (or misinformation/biased information) about their technologies; thus, the cost of acquiring the right information becomes higher. Ports, therefore, become wary about the information on technologies even though they fit in properly. If ports have the right information on measures, the risk of not adopting technologies may be reduced.

**Adverse selection**

Selection of technologies (measures) by investors (ports) may happen on the basis of visible aspects such as price rather than performance (e.g., abatement potential), owing to the fact that the supplier of technology (vendor) knows more about the technology than the buyer (port). Additionally, the supplier may not relay the information to the other side accurately. Thus, adverse selection is a moral hazard in that actors (ports) may behave opportunistically [75]. This hinders investment or makes the investor select non-beneficial technology or services.

**Principle agent relationship (problem)**

The principal agent contractual relationship is not always efficient. The principal (port authority/landlord) may invest in decarbonisation but cannot see what the agent (port operator/energy provider) is doing. Thus, the principal strictly controls and monitors the agent's decarbonisation management (i.e., check out the value of investment, requiring short payback rates or high hurdle rates, etc.). This leads to missing opportunities for

implementation of better measures in that the agent avoids implementation and even further investment. It is thus important that the agent provides the principal with reliable, accurate, and publicly available information on decarbonisation measures implementation, while the principal needs to streamline and simplify the relationship to improve sustainable performance.

**Split incentive**

Split incentive occurs when two parties have different goals (goal conflict) and information (asymmetric information), in addition to the risk bearing costs (cost not shared) such as the case in a landlord-tenant relationship. So, if the costs of investment in decarbonisation measures do not yield benefit to the investor, i.e., measures' cost is borne by one party while the other enjoys the benefit, this demotivates and decelerates the adoption of the measures. Ports may invest in decarbonisation technologies, but those who run the technology may benefit through energy efficiency or avoiding paying emission tax (regulation compliance) while not bearing the costs or sharing the benefits. This happens between the port authority (landlord) and tenants (private terminal operators, industries), concessionaires, lessees, and other parties under contracts.

From another angle, this also includes land transport and ships, as they are the customers of the port. Decarbonization costs ports a fortune, particularly in technologies for decarbonising the ship port interface, such as onshore power supply (OPS), alternative fuel bunkering, pre-booking, just in time (JIT), and virtual arrival [76–78]. While ship owners, charterers, and even cargo owners gain benefits, i.e., profit by less turnaround time and improving their green image by less carbon foot printing, this leaves ports distanced in implementation. The same is true regarding ports and other subcontractors, for instance the difficulties of energy efficiency management. The landlord, who is legally bound by the government's regulation to decarbonise, manages his own limited operation and superstructure contrary to subcontractors that have no incentives to manage their operation (energy efficiently) due to their focus on profit, besides the high cost of management (costs of lights and generators renovation and retrofitting including heating and cooling systems, etc.). If ports implement decarbonization measures, the cost is born by ports while subcontractors are free riders (beneficiaries with no costs).

## 4.2. Economic Non-Market Failure

**High costs and access capital issues**

The costs of decarbonisation technologies are high, including those for port operation (electrification of cargo handling equipment) or ships (OPS). Retrofitting one berth with OPS costs USD 1 million [78]. The same is true regarding the high cost of investing in LNG and alternative fuel bunkering infrastructure. There is an imbalance between environmental quality and economic feasibility [79]. Port authorities are aware that an environmental policy generates additional costs and expenses; while they recognise the necessity of it, they attempt to reduce the level of policy intensity or delay its enforcement as much as they can [23]. While the costs are extremely high, lack of capital was also identified as a factor that restricts port sustainability implementation [80], exacerbated by the lack of subsidies and fiscal regimes that vary too much from country to country. Therefore, restricted access to capital and its high cost (interest rate) is considered to be a key barrier to investment in decarbonisation technologies. The latter is worse in developing countries, where the cost of capital for renewable energy is still very high.

**Hidden costs**

The life cycle costs, transactional costs, and commissioning, operation, and maintenance or disruption and other overhead costs (e.g., research, consultancy, feasibility and appraisal studies, procurement, monitoring, data handling, retrofitting, training, commissioning and decommissioning (e.g., large wind generators) may increase the hidden costs and thus hold back deployment of technologies.

**Risks**

Investment may be seen as a risk, by risk avert ports, particularly if the payback period is long whilst the technology depreciates fast, and subsidies or grants are unavailable. As such, implementation does not achieve an appropriate return on investment. Another business risk may be seen from the perspective of energy supply (LNG) and security (sanctions) that was compromised during the recent crises in Ukraine/Russia and the Pandemic. On the other hand, the risk of stranded assets and asset specificity are also barriers [75]. Specificity means that assets are specific for maritime operation and cannot be used outside this domain (port). For example, in relation to the OPS for cruise terminals, not only does seasonality affect this, but also no other ships can use it including other freight transport modes (trains and trucks). On the other hand, stranded assets emerge when a technology becomes non-operational, yielding no benefit, due to its non-usability. For example, LNG bunkering and equipment that is run on LNG will not be used in the future as other zero or near zero emission alternative fuels should be used to reach the climate goal.

**Heterogeneity**

Decarbonisation technologies and measures may not be cost effective (efficient) for some ports, including some other polluters (ships, land transport), due to heterogeneity in ports, and its operations and customers. This is to say that ports have different conditions, including governance, business models, geography, throughput, cargo specialisation (container, passenger, RoRo, dry and liquid bulk, and general cargo ships), trade, geographical location, and energy supply and utility. It is worth noting that multi-port stakeholders (government, city, service providers, customers, suppliers, etc.) involved in decarbonisation are also heterogeneous with varying interest and stakes. Satisfying and meeting the expectations of all stakeholders is a complex issue. Generally, what can be applied successfully in one port may not necessarily be successful if implemented in another port. On the other hand, port heterogeneity generates nonuniform responses and complicates the whole implementation process. Furthermore, many ports are considered small vis a vis large ports that have the potential to implement sustainability, such as port differentiated port dues [81]. Small ports find it difficult to sustain profitability once large costs are incurred in decarbonisation implementation (e.g., the case of ISO certifications [82]).

*4.3. Behavioural Barriers*

**Form of information**

The information on decarbonisation's benefits, abatement potential, return on investment (ROI), and costs is vital to mitigate barriers to implementation. Many ports avoid implementation if they doubt the benefits of technologies. Furthermore, the exchange of requested information among stakeholders, e.g., between ships and ports or port authorities and terminal operators (the case of energy consumption, or carbon foot printing), is not optimal or accurate. This is attributed to confidentiality and reluctancy in data sharing, or just to avoid the high cost of carbon reduction [83,84]. Therefore, the form of information provided should be Specific, Vivid, Simple, and Personal (SVSP).

**Credibility and trust**

The lack of credibility and trustworthiness in the technology and service provider (supplier, seller, or vendor) and their information may lead to improper and inefficient choice of decarbonisation measures (technologies).

**Values**

Implementation of decarbonisation measures is influenced by **norms and values** of polluting companies/organisation (ports) [24]. Not having concerns and moral commitment to the environment and society, in addition to lack of real ambition, particularly from top management, influences the adoption of decarbonisation measures. Lack of commitment and disposition of port managers was identified as a barrier to decarbonisation implementation [85].

**Inertia**

In general, psychological and cultural aspects can also represent challenges for industries' energy transition [86]. Resistance, not welcoming change in the work environment, leads to avoiding and ignoring the efforts to decarbonise the port even when cost-effective measures can be implemented. As decarbonisation is an emerging concept, implementation may face strict inertia, particularly by top port management, including other stakeholders, because it is costly and may divert them from conducting profitable business simply and with no complications, i.e., implementation is an extra burden [87]. Inertia also includes port labour that may favour old Cargo Handling Equipment (CHE) and operations over new equipment that requires new knowledge and training. Another source of inertia may emerge from port surroundings (communities), city boards, NGOs, and other stakeholders who contest (socially acceptance issue) decarbonisation technologies that compromise city safety (e.g., ammonia and hydrogen storage and regasification platforms) and energy intensive technologies (e.g., electrification).

**Bounded rationality**

Instead of making rational decisions based on correct information, ports (decision makers) are hooked by bounded rationality; thus, they use the rule of thumb (non-rational decisions) in their selection of decarbonisation measures [75]. This is attributed to their lack of expertise and the inability to assess the life cycle costs of measures and conduct proper investment appraisals, while also needing to shorten and decrease the time and cost of technology implementation.

*4.4. Organizational Barriers*

**Power**

When environmental managers in ports lack power and status, the decarbonisation issue has less priority in decision making. This is linked with port institutional power, which is different based on their governance [88]. Depending on the role of the port authority, implementation of environmental measures differs. In the Hanseatic tradition, e.g., in North Baltic countries, ports are decentralized, and the local government or municipality consequently has strong control over the port. Hence, the port authority has more power and autonomy to implement environmental measures without complications. In contrast, in the Latin tradition, national government has strong control over the port, and thus the port authority has less power and autonomy in taking environmental actions [44,89].

**Culture**

When the culture of the ports (polluters' organisation) is developed based on respect and appreciation of the environment and effective Corporate Social Responsibility (CSR), decarbonisation issues are always promoted, and thus managers are encouraged to invest in decarbonisation, while individuals (employees) also take actions to decarbonise. This is not the case in many ports though.

*4.5. Institutional Barriers*

Institutional barriers are issues caused by political institutions, i.e., state government and local authorities [68]. Major institutional barriers for seaport sustainability are related to supply chain issues that require intersectoral and interjurisdictional collaboration and multimodal integration [79,88]. The institutional structures of firms form their responses to technological opportunities and policies [67]. The following are various institutional barriers:

**Political roles**

Overlapping port governance may undermine the port's political role in making decisions toward decarbonisation. Overlapping governance is linked to institutional forces [88]. The unnecessary fragmentation, complexity, and red tape of the multi-level port governance scene definitely decelerates the implementation of environmental measures [90,91]. Zooming out, different political systems between countries and regions may influence the transportation of electricity or fuels, which in turn influence safety and security [86].

**Governmental regulations**

There is a lack of strict decarbonisation regulations, making ports uncertain about investing in technologies. Absence of regulation, policy, and managerial key performance indicators limits port sustainability performance [92], which leads to uncertainty in investment [86]. Even if some regulations are in place, they are not strictly enforced by national and local government or port authorities [84,93]. With no effective regulations, ports may not implement technologies because there is no accountability, and the cost benefit ratio is therefore not above unity. This being so, implementation of strict regulation yields an unlevel playing field that favours ports that do not implement regulation. Overall, existing regulatory and fiscal systems are not helpful; for example, tariff structure do not reward ports for investment in renewable energy. Some regulations may even hinder port investment if they do not allow operation of certain technologies. For example, ports that want to invest in renewable energy (wind or solar) may be hampered by regulations that ban electricity trading (dumping of access energy to the national grid for later reclamation). Similarly, some municipalities may ban the storage of alternative fuels (ammonia, hydrogen) that risk port and city safety. Some cities may even ban the OPS due to the high load on the already weak local grids. This is attributed to lack of codes, standards, local or international regulations, and guidelines for alternative fuels bunkering (e.g., LNG and methanol, hydrogen, and ammonia) or the carriage of alternative fuels by ships.

**Industrial norms and mimetic actions**

The norms of ports, for example in training and improving their managers and employee's awareness and technical skills, are important for better implementation of decarbonisation. Neighbouring ports also play critical roles in port decarbonisation through mimetic and normative forces. Ports that have limited norms pay less attention to environmental issues. Thus, neighbouring ports can mimic such norms leading to similar approaches that do not improve decarbonisation implementation.

*4.6. Technological Barriers*

**Incompatibility**

There is incompatibility between decarbonisation technologies and port types and operations. Some technologies suit container terminals, whereas bulk or general cargo terminals may have difficulties in implementing the same technology. An example is the OPS barriers (see [94]) in that bulk ships are rarely fitted with OPS vis a vis liners (passenger or containers) that visit ports very often and may take advantage of ports' OPS. Additionally, some technologies are not compatible with existing systems, which need separate systems (fuels, and hybrid, electrified, automated and non-automated equipment, etc.). Similarly, there is an issue concerning port OPS and charging stations when their voltage and frequency is not compatible with ships [95]. Many ports lack physical spaces for new technologies. For example, bunkering storage with safety zones, OPS, charging stations, and new maintenance stations, require new and large expansions that many ports do not have space for.

**Interference with ports' main processes**

Some technologies may interfere with port operations and create issues. For instance, a fault in a smart grid may lead to corruption of perishables or logistics inefficiency, making the port accountable. Additionally, port decarbonisation may depend highly on electrification. Thus, peak load may demand more electricity than is available or based on renewables. On the other hand, digitalisation (information communication technology) triggers cyber security issues, which may leak port information, compromising privacy and competitiveness.

**The complexity of measures**

There are various complexities in technology implementation, which result in reluctance in its initiation. Some technologies require new infrastructure or long renovation or construction times, such as large fuel storage facilities, OPS infrastructure, and large batteries that take space and thus may hinder some operation. OPS is generally influenced

by the source of port electricity (green or otherwise), type and frequency of calling ships, and distance to the city grids. The time taken to connect and disconnect the OPS, or charging, is another issue, particularly for ships with limited berth windows (RoRo, container, passenger and ferries); connection and disconnection of OPS may take up to one hour, which may increase a ship's turnaround time [78]. The same is true regarding limited national grid capacity that cannot feed the port OPS.

**Technology readiness and abatement potential**

While some decarbonisation technologies are available, some may not give high outputs (e.g., renewable energy generation through tide or wave energy) or abatement potential (e.g., pre and after treatment technologies for cargo handling equipment). Ports that are not sure about the abatement (not being at least verified) would not invest in the technology. On the other hand, although some alternative fuels, including biofuels, are being used in the market, the availability in large scale is still limited [96]. Ports may need to compete with other industries and sectors to get access to such fuels. Furthermore, some technologies, such as carbon capture and storage or utilisation, are not yet mature. The latter does not even have a potential market thus far.

### 4.7. Time Barriers

Generally, decision making takes time and deploying measures also takes time. Also, some technology implementation decisions may need to go up to local, national, and federal governments and diverse ministries and organisations at different hierarchy levels [87]. This behaviour leads to long lead times for taking decisions (or to outdated decisions being made). On the other hand, some ports may already have long-term agendas and strategic plans that do not count decarbonisation as a priority.

### 4.8. Administrative Barriers

The administrative barrier occurs when a port lacks the awareness, guidelines, resources, experienced staff, and technical skills to analyse, make decisions, and oversee the implementation of decarbonisation solutions [84,97–99]. Additionally, there is a large number of diverse small ports (companies) that may not have the management expertise needed to evaluate and implement decarbonisation solutions. This is further complicated when ports use third-party services to conduct some business (such as operators, 3PL, etc.). In this case, ports as landlords may be removed from day-to-day operational issues leading to lack of ability to administrate decarbonisation. On the other hand, there exist administrative conflicts inside the port different department. For example, a finance department may favour low capital cost, procurement may favour heavy duty and operationally efficient technologies, while environment and energy departments may focus on lower abatement potential and greater energy efficient technologies.

### 5. Solutions to Implementation Barriers

The most important solutions to port decarbonisation barriers are embedded in policy and management instruments and tools. These tools are widely used in environmental measures implementation [44,100,101] and can be implemented by public authorities (municipal, central, and federal government) and/or port authorities [37]. Alamoush et al. suggested different policy and management tools that can be used to expedite port decarbonisation (i.e., regulation and standards, incentives, disincentives, compulsory agreement, voluntary agreement, capacity building, information sharing and raising awareness, strategic plans, inventory, monitoring and reporting, and the three Cs (cooperation, coordination, collaboration)) [37]. Additionally, based on the broad previous analysis, other solutions are proposed (i.e., investment and finance pull, managerial development, and miscellaneous solutions). These tools provide useful help in decarbonising ports and extending beyond ports' supply chains [28]. Overall, these solutions influence port authorities and other port polluters, including private port operators (port operation), ships (ship port interface), land transport operators (land transport port interface), in addition to industrial activities and

domestic shipping, inland waterways, and marine services. In other words, they work as drivers for carbon polluters that increase their uptake of decarbonisation technologies and measures. As per the typologies in Figure 1, the following subsections explain these aggregated solutions.

### 5.1. Regulations and Standards

Regulations are commonly managed by public authorities (e.g., national governments, municipalities) due to their jurisdiction over port authorities [37,102]. The **government** regulatory role is detrimental to empowering ports [79]. PPAs need to implement the provisions of the Paris Agreement (climate change mitigation) and also domesticate environmental regulations (conventions), e.g., IMO International Convention for the Prevention of Pollution from Ships (MARPOL), and make their implementation by port operators, tenants, ships (in ports), and land transport legally binding [17,79,103,104]. In this sense, national environmental law is relevant to port authorities [79]. Ports authorities are bound by such regulations, particularly if they are the operators (a polluter). Additionally, port authorities set up some roles and regulations for decarbonisation as a response to the upper regulatory pressure. A case in point: port authorities use decarbonisation regulations and liability standards to ban fossil-fuels-run cargo handling equipment, and polluters (e.g., private port operator) may be required to use specific technical measures to reduce and control emissions [44]. Regulations, policies, and liability standards are necessary when developing environmental and energy efficient ports, and their absence will restrict port sustainability performance [92,102]. This mitigates the current loosely enforced sustainability rules in port regions (lack of strict regulations) that hinder sustainability implementation [84]. All in all, regulations are vital for port sustainability implementation and energy efficiency [92]. It is worth noting that many ports require government interventions through policies and instruments that help ports to bring about innovative actions such as decarbonisation [24,87], for example by instituting supportive policies to encourage the large-scale production of zero-carbon fuels. Of consideration, ports should have a decarbonisation policy (strategy) that has pathways and targets to meet the climate goal.

### 5.2. Incentives and Economic Support

As a major barrier, the **lack of capital and financial resources**, high operational costs, tight budgetary accounts, and the poor financial assistance from the government or relevant authorities (financing) are recognised as predominant challenges that constrain port management from sustainability implementation [80,84,85,105]. To add to this challenge, port managers have put a large emphasis on economic sustainability over environmental sustainability, i.e., they excessively focus on short-term economic goals rather than long-term sustainability advantages, as revealed by Becker and Caldwell [106]. Therefore, incentives, environmentally differentiated port fees, grants and subsidies, and tax exemptions, among others, should be utilised to stimulate decarbonisation.

While port authorities incentivise tenants and operators locally, they also incentivise ships based on environmental indices and certifications that partially target $CO_2$ reduction, for example: environmental shipping index (ESI), clean shipping index (CSI), green award (GA), and GHG emission rating (GHG ER) [28]. In addition, vessel speed reduction (VSR), the use of Onshore Power Supply (OPS) by ships and ports, and ports' energy audits and utilisation of renewable energies, virtual and just in time arrival, and adoption of other abatement technologies are also incentivised through monetary rebates, deductions, tax exemptions, and berth priorities [28,79,107,108]. Ports also incentivise hinterland transport (trucks and railways), e.g., the case of Hamburg Port incentives for the railways and locomotives, and Long Beach and Los Angeles port authorities clean truck programme [27,28,36,37].

Incentives are commonly provided by public authorities (governments, municipalities) because of the ability to fund such programmes, particularly when the port is a public model (revenues are publicly managed) [37]. The different forms of incentives assist ports, ships, and land transport operators to offset the high cost (capital and operational costs) of

decarbonisation measures [24,44,93,95,109], thereby mitigating the financial barriers [80,84,85]. Thus, it is recommended that government (public authorities) provide initial funding and launch publicly funded development programmes to lower the cost and risk of starting innovation projects in ports [110].

### 5.3. Disincentives and Punitive

PPAs need to charge polluters (the polluter pays principle), change the tariff and prepare different rates (price increase/pricing), and utilise the emission trading scheme or carbon tax and other market based measures (MBMs) to drive ports' tenants and operators, ships, and land transport to increase energy efficiency and reduce their carbon emissions, i.e., uptake of decarbonisation measures [37,111,112]. Disincentives create funds to invest in cleaner technologies and manage and offset GHG emissions elsewhere [37]. For example, to decrease idling emissions and congestions, the Port of Long Beach (POLB) and Los Angeles truck PierPass programme (This measure reduces trucks congestion, idling thus its $CO_2$ emissions in ports) charges extra fees (USD 50) for handling containers in peak hours [113]. While market based measures are used locally in some countries, the European Union. advanced in its construction (regional), and MBM are currently discussed by the IMO (GHG strategy) for global enforcement [44,114–117]. Use of a disincentive is very important for port decarbonisation, especially when considering that port polluters will always try to escape paying the disincentive by adopting low carbon emission measures [24,37,118].

### 5.4. Voluntary Agreements

Ports may sign voluntary agreements (no legal obligations or compensations) with polluters ($CO_2$ emitters) to minimize their emissions. This is deemed to be a successful way of achieving carbon reduction [78], as it increases the uptake of decarbonisation measures by polluters voluntarily, with no costs for ports. In light of this, PPAs sign voluntary agreements with port, ship, and truck operators to reduce carbon emission in ports. Examples include POLB's green flag environmental programme for voluntary speed reduction, which reduced 26,000 tonnes of $CO_2$ between 2002 and 2007 [107], Rotterdam PA's modal shift program, which was signed with port and truck operators, and the California state authority's program for replacement of old trucks [79,119,120]. It is worth noting that ports, ships, and land transport operators who sign such agreements improve their green image and receive recognitions and awards in addition to privileges such as priority berthing or fewer inspections [37].

### 5.5. Compulsory Agreements

PPAs sign compulsory agreements with port polluters that bind them decarbonise. For example, in concession agreements or leases, designed by PPAs, decarbonisation of port concessionaires, particularly private operators including tenants and industrial firms, can be considered [79,93,121]. PPAs thus need to negotiate the inclusion of technical terms that require port operators and tenants to decarbonise [17,122,123]. The same is applicable when signing leases and agreements and issuing 'licence to operate' certification to trucking companies, dredgers, and barges, among others, where they are required to be environmentally friendly and have decarbonisation agendas. Evidence in [124] indicates that PPAs can include terms in their contracts with port and truck operators to facilitate a modal spilt while, at the same time, negotiating consequences and solutions. In another case, the Tokyo Port Terminal Corporation includes GHG emission reduction terms in the port operators' concession agreements [125]. On this basis, using compulsory agreements with polluters influences the uptake of decarbonisation measures.

### 5.6. Capacity Building (Training, Awareness Raising, and Technical Support)

Capacity building programs (technical support, training, and awareness raising) by PPAs can target ports and other polluters to help them reduce their carbon footprint. While the lack of expertise is considered a major challenge in achieving sustainable development

in ports [80], the lack of competence and technical skills (capacities) is also identified as a barrier that impedes the adoption of decarbonisation measures (including cost-free operational measures) in maritime transport, including ports [66,73,97,98]. Skilled employees are inadequate in numerous ports, even in developed country ports [105], and they are strongly required to efficiently operate decarbonisation and digital technologies in maritime transport [126].

**Capacity building** is one of the key solutions that contributes to sustainability implementation in ports (including climate change mitigation) [27,127]. Therefore, to improve the expertise and raise the awareness of port polluters, and to mitigate the above-mentioned barriers, PPAs implement capacity building programs through **educating and training** tenants, stakeholders, and employees in order to reduce their carbon footprint [17,128,129]. Awareness and understanding of the depth and context of maritime transport sustainability pathways plays a pivotal role in the implementation [66,130].

Importantly, training and reskilling are required for a just and streamlined transition (qualifications), which includes empowerment (training and certification) of employees and machines operators, particularly on the shop floor [131], which ultimately improve their port green behaviour [132]. Training needs to include handling of alternative fuels (safety and security). Similarly, ports should train the land transport (trucks) for eco and energy efficient driving [133]. Ports should also encourage employees and tenants to use car and van pooling, public transportation, and cycling [19,134]. Additionally, PPAs may act as community and port cluster managers where they can engage with different stakeholders, and invest in facilitating activities such as information technologies training and education and promotion and marketing of the port [135]. In the same vein, PPAs can support and open the door to researchers and technologists to develop technical and commercial viability of the most promising decarbonization measures, allowing technology verification while working as testing beds.

Considering port cyber security issues, leaks of information due to dependency on information technology (IT) and digitalisation and the risk of privacy disclosure are fundamental technical barriers to the adoption of decarbonisation measures (energy efficiency). PPAs can mitigate such barriers by providing **technical support** for port operators and other polluters [102]. Other technical support for ports, including other polluters, contain technologies for monitoring emissions and demonstration projects [102]. Evidence from USA and Canadian ports indicates that sustainability awareness and training programs have positive effects on port sustainability implementation [88,102], including GHG emission reduction [136]. Finally, while developed countries are advancing in decarbonisation implementation [36,37], it is recommended that ports in developing countries are not left behind, i.e., aids, technology transfer, and capacity building should be provided to them by front runners to help them to face the decarbonization process.

*5.7. Information and Knowledge Sharing Incubators*

Many ports lack systematically collected and necessary data to track the evolution of technologies and costs, resulting in misconceptions and out-of-date data that undermine the decarbonisation effort's effectiveness. Lack of information on energy efficiency measures benefits, including costs, is a key barrier to implementation of decarbonisation measures (energy efficiency gap) in the maritime sector [43]. The low perceived benefits of decarbonisation technology may decelerate the polluters' adoption of measures [44]. Therefore, sharing information about decarbonisation is a vital step to mobilise port polluters' attention toward decarbonisation [37].

PPAs can serve as a central point of **knowledge** and thus share and disseminate information about how polluters can decarbonize, including providing feasibility of adoption of advanced decarbonisation technologies and measures [37,131,137]. PPAs, acting as community manager, have an entrepreneurial role, particularly once they share information with a variety of stakeholders [91,135]. For example, they can share with port operators the emission inventories guidelines, methodologies and results, decarbonisation

best practices, and information on technologies and measures abatement potential [44]. Importantly, developed ports (advanced in decarbonisation) need to share information with other neighbouring and regional ports [35,91].

Of consideration, considering that many ports lack specific decarbonisation expertise, they improve their experience when benefiting from those specialised in the fields of energy management (e.g., ISO certifiers) and decarbonisation. This includes getting knowledge and support from consultancy, energy management, technologies, and manufacturing companies, in addition to research and development companies and universities. Ports can also be innovation drivers in port clusters by housing knowledge hubs, incubators, smart labs for start-ups and scale-ups, and even software and hardware solutions that can help other ports or the whole maritime industry. This can materialise through cooperation (agreements) with universities and research institutions, and creating a good business environment for research and development (R&D) focused firms, research centers, consultancy firms, and start-ups [24].

Within this context, ports can understand decarbonisation practices and develop inventory for their energy consumption (emissions) [24]. In so doing, data and information collection and sharing promote better understanding of decarbonisation measures, may reduce the costs associated with implementation, and demonstrate the economic and societal advantage of technologies, which, consequently, enhance decarbonisation implementation. While information can be shared through workshops, conferences, technical reports, and sustainability reports, information can also be shared via formal conversations, face to face meetings, telephone conversations, e-mail, SMS, banners, and social media platforms such as WeChat, Line, Twitter, and Facebook [132]. Digitalisation, however, remains the catalyst for information sharing. Investment in digitalisation (IT and ICT) not only improves port efficiency, monitoring and reporting of emissions, and data collection, but also improves and expedites information sharing among various stakeholders. For example, the IMO suggested, through digital platforms, giving port stakeholders (charterers, shipowners and operators) access to port data in addition to terminal databases [83]. The same is true regarding port traffic management that depends on digital information sharing; it also reduces ships' emissions in ports (turnaround time reduction). Digitalisation, through electronic data exchange, port Collaborative Decision-Making (PortCDM), and other digital technologies [138], facilitates ships' virtual arrival, Just-In-Time (JIT) berthing, and Vessel Speed Reduction (VSR) [28,93,139–141].

### 5.8. Port Strategic Plans

Ports' strategic and long-term development planning (master plans) encompass various future port activities. While strategic planning targets optimising port operations and activities [142], it can also be utilised to improve port sustainability as it holds enormous potential for making port operations more eco-efficient, particularly when the environmental aspects are considered. Ports can indeed plan for decarbonisation and environmental sustainability targets as part of such strategic planning [21,37,79,143]. Ports, in addition, can plan for all the other implementation policies, tools, and instruments to develop and transit themselves in a sustainable direction [37]. The plans include, but are not limited to, designing of green and sustainable ports [17,20], environmental and energy management systems [79,87,103], reduction of shipping emissions in ports [78], and $CO_2$ emission reduction in expansion projects (e.g., port of Rotterdam expansion project Maasvlakte II) [9]. Many EU ports plan for OPS and alternative fuel bunkering [36]. In the USA, San Pedro Bay Ports planed for replacement of older cargo handling equipment with new cleaner engines over a specific period [144], while Ports of Los Angeles and Long Beach planned for zero-emission ports, which, as investigated by [102], would decarbonise the ports by 2035.

### 5.9. Inventory, Monitoring, and Reporting

With respect to port operation, emission, and energy inventory, monitoring and reporting are a key and primary course of action. The inventory is the most important step in

decarbonisation, as we cannot manage what we cannot measure. Inventory helps ports establish an emission baseline and to identify effective and appropriate measures to successfully implement decarbonisation measures and investment strategies. Inventory advocates collection of emission data while monitoring, reporting, and tracking enable improvement and benchmark emissions over the years [36,131,145–147]. The world port climate initiative (WPCI) segmented port GHG emission into three scopes, which incorporated emission and energy consumption (electricity) of port authorities, including those of tenants, and the port, ship, and land transport operators [148]. Notably, a standardized tool for port carbon footprint using the WPCI and the IPCC guidelines and the GHG protocol was built [149]. Many developed countries' ports establish broad sustainability reporting, including the European Seaport Organisation (ESPO). These reports enable and work as official statements of port emissions while at the same time greening the port image. Within this context, it was revealed that sustainability reporting has a positive effect on port sustainability implementation [88,102,136]. For port reporting, it can be suggested that ports dedicate an environmental section on their websites specifically to disseminate information on port energy efficiency and decarbonisation efforts and results. This raises the awareness of the industry while improving green image and competitiveness.

*5.10. Investment and Finance Pull*

PPAs are recommended to seek investment and green finance opportunities to start the transition process and mitigate the financial consequences. Different finance taxonomies and capital funding that support climate change mitigation are available for ports. Unlocking such funds and getting access to them greatly helps ports to afford decarbonisation costs. In this sense, PPAs need to claim or solicit public funds to finance decarbonisation from, for example, the European Maritime Climate Fund, the Getting Zero Coalition (GZC), and the Financing Sustainable Maritime Transport (FIN-SMART) forum. Many commercial banks offer green loans; thus, ports should utilise this opportunity. PPAs need to collaborate with financial institutions, such as domestic, international and regional development banks, funds, and investors, to invest in decarbonisation technologies, in addition to putting up custom-made financial mechanisms, including low-interest loans. It has been highlighted that access to capital in the future may, to a large extent, depend on how sustainable the targets that require financing are, while banks and lenders may need to report their sustainable and green investments [28].

Equally important, considering that many countries still lack funds, particularly developing countries, least-developed countries, and small island developing states, revenues from the IMO and EU proposed Market Based Measures (MBMs) (fuel or emissions tax) should assist these countries' ports in funding decarbonisation. The carbon offset market (MBMs) is expected to grow into a multibillion-dollar industry over the next decade [150]. The revenues not only bring advantages to shipping but also to others, such as ports and national governments.

In terms of investments, port authorities or operators may invest in transport or transshipment companies operating around the port (railway transport companies or intermodal terminals and dry ports) by taking over their shares, e.g., port authority of Hamburg HHLA share in intermodal operator—Polzug, and Barcelona investment in intermodal terminals and dry ports in the hinterland in Zaragoza, Toulouse, Perpignan, and Madrid cities [7]. This improves port connectedness with the hinterland and increases its share and control over environmentally friendly transport modes. As with the private terminal operators (build, operate, transfer (BOT)), particularly those run by shipping companies (liners), ports need to attract investment in decarbonisation technologies for shipping and land transport companies (Public Private Partnership). Examples include investment in shoreside electricity, either OPS or charging stations, and digitalisation of gates and operations, including the single window and port community systems, to be used by the investors' vessels, trucks, or railways, etc. Another type of investment can be sought from mega shippers and retailers or cargo owners (such as Amazon, and Walmart

in the USA) to support port decarbonisation. It is worth noting that a high inclination to invest in renewable energy technologies is seen in industrialised countries, in contrast to developing countries where capital is scarce and interest rates are high. While this demotivates investors (ports), there is a need for large global investments in developing countries ports to help in decarbonisation (e.g., through renewable energy) [86].

*5.11. Managerial Development*

Awareness of port managers about environmental benefits improves sustainability implementation [84]. In this regard, PPAs need to take actions to develop and improve managerial aspects. Hence, it is suggested that ports need to boost managers' commitment and divert their disposition and commitment to the environmental targets in general, and particularly to decarbonisation. Port managers' positive orientation was identified as a driver of sustainable supply chains [151], while their commitment to environmental actions is also considered as an influencing factor to greening ports [152] and land transport [133]. Consequently, this increases the port compliance with environmental laws and regulations [153].

Commitment and disposition of managers can materialise once ports improve their environmental knowledge through training and specific courses, or when hiring environmentally aware managers (bonus). In general, improving the port commitment positively influences managerial commitment. Hence, ports can demonstrate their commitment to decarbonization by setting ambitious targets. In fact, the port commitment must be established and demonstrated from top to bottom and must also be manifested by goals set by PPAs. For example, the ports of Rotterdam, Antwerp, and Singapore revealed their commitment to become zero carbon ports by 2050. In addition, ports improve their commitment through implementing corporate social responsibility (CSR) (Corporate social reporting (CSR) is concerned with learning about the effect an organization has on society and allows the organization to be accountable for these responsibilities [24]) strategies [24].

*5.12. Cooperation, Collaboration, and Coordination (3Cs)*

Port governance in the first place necessitates ***engagement of a variety of governing bodies*** at multi-levels, e.g., ministries of environment, transport, trade, and maritime, customs, city board, and municipality [37]. The same is true regarding governance of decarbonisation. The presence of multiple institutions in port governance at diverse tiers (i.e., national, regional, and local) increases the bureaucracy in decision making [90]. Therefore, intersectoral and interjurisdictional (multi governing bodies) cooperation, collaboration, and coordination are a stimulus for ports' sustainability implementation [88,91,121].

Weak collaboration with shipping lines and other policy makers is identified as a barrier to port sustainability [84]. Thus, another form of cooperation to achieve green targets is with ***different maritime stakeholders***, i.e., terminals and port operators, neighbouring and regional ports, shippers, shipping lines, city (Port-city), and other land transport (hinterland trucks and railways), [37,152,154–156]. For example, a case study of California ports revealed that achieving zero-emissions CHE is feasible when there is stronger collaboration with main stakeholders (e.g., port authorities, the government, industries, and community groups) [102]. The 3Cs between ports and other stakeholders and governing bodies decrease market constraints (competition) that undermine sustainability implementation [85], and establish level playing fields. Additionally, the 3Cs assist in setting performance standards and pricings, and facilitate making uniform the incentives and disincentives locally and regionally, which eventually protects competition through cooperation (coopetition) and minimises free riding and spell over effects [37,85,157]. In the same vein, cooperation and collaboration with the universities, academia, R&D centers, and consultancy firms are also important to feed the ports with state-of-the-art technological solutions while enhancing efficiency and productivity.

Another form of interorganisational integration (3Cs) is ***building and joining networks, alliances, collations and relationships*** at national, regional, and international levels. These

inter-organizational networks improve sustainability implementation through sharing of resources, skills, expert consultation, best practices, and lobbying power [158]. To mitigate higher implementation complexity, it is suggested that ports involve themselves in lobbyism and alliances with regional bodies or in the IMO through their membership of the International Association of Ports and Harbors (IAPH) [93]. Similarly, ports can increase their links and coordinations, and gain industry association support through, inter alia, the engagement in the forty cities for climate actions (C40), local governments for sustainability (ICLEI), Eco Partnership, or the European Sea Port Organization (ESPO), including the ECOPORT initiative. Even formation of local associations (e.g., the British Ports Association) is helpful to address the concerns of port authorities and private port operators or companies [105].

Overall, deficient collaboration among policymakers causes implementation to deteriorate, while efficient collaboration improves ports partnerships and the implementation of the SDG Goal 17 (partnerships) [84]. When a port makes efforts in isolation, it is likely that contradictions with other organisations will occur; therefore, undisputed perception among policymakers, stakeholders, organisations, and institutions is a necessity for realizing port decarbonisation.

### 5.13. Miscellaneous Solutions

Ports need to conduct **feasibility studies** (technical and cost benefit ratio analysis) in addition to life cycle costing that show carbon footprints (environmental impact assessment). This decreases the investment risks and shows the best value of technologies. Future alternative fuels (ammonia, hydrogen, LNG, methanol, etc.) entail risks and hazards (safety issues), such as fire, explosions, and poisonous gasses; therefore, **safety guidelines** should be fashioned and utilised to mitigate such risks.

Due to **scarcity** in port spaces and the limited opportunity to expand, ports of the future may need to relocate operations and activities to outside the city where other pollution activities (air, noise, visual, etc.) are away from residents.

**Certifications and audit** are also vital steps ports need to take, e.g., the ISO 50001 [54] energy management systems, ISO14001 [159] environmental management system, the European Union's eco-management and audit scheme (EMAS), and ESPO self-diagnosis method (SDM), and port environmental review system (PERS) [27]. Certification and audits would advance port energy efficiency and help ports detect issues with energy management while, at the same time, keeping up the monitoring and review cycle to maintain efficient performance. Within this framework, ports can establish KPIs to reduce emissions and energy consumption.

**Stakeholder** mapping and management is an essential step in port energy transition and decarbonisation. While the stakeholders' relationship with and importance to decarbonisation differ appreciably, stakeholders also have contradictory stakes and priorities, resources and competing interests, and should thus be prioritized and managed appropriately. By this means, stakeholder management nurtures trust and collaboration with the port, leading to better decision-making, resource allocation, and innovations that boost decarbonisation efforts. Additionally, stakeholders' engagement can help mitigate the split incentive issue; thus, ports should engage key stakeholders in the decarbonisation process (procurement, purchase, construction, and operations), in that ships, land transport, terminal operators, and different departments of the port authority, including employees, become part of the decision making and operational and financial accountability. The involvement of cargo owners can ensure great leverage to port environmental programmes [160]. Within this context, ports can also promote industrial ecology and the circular economy due to the presence of various stakeholders in the port (industries, tenants, city, logistics chains) [27].

While the current business models may partially support port decarbonisation, particularly the landlord model, **new business models** should be introduced. This includes, but is not limited to, third party investors in technology solutions. Another model can be costs divisions (sharing) among beneficiaries. Considering that there are challenges related to the

investment cost, both for terminals and ships, the division of these costs between different stakeholders (shipping line, terminal operator, and port authority) help in reducing the total costs, and thus assists in shifting from cheap bunker fuel, while at the same time obtain environmental benefits [24].

## 6. Opportunities of Port Decarbonisation

The previous sections have shown how the decarbonisation process is fraught with challenges, problems, and pitfalls. Therefore, many solutions have been proposed to ensure effective, safe, equal, and fair transition. However, decarbonisation should not be viewed as a risky transition, but rather as creating opportunities to develop ports by opening new doors toward expansion, growth, entrepreneurship, new horizons, and integration into green supply chains. These opportunities can be summarized in the following points.

- Decarbonisation can be an approach that enhances the port's competitiveness because the adoption of decarbonisation puts ports on a sustainable track (green image), which would attract environmentally friendly shipping lines and customers who prefer ports with low rather than high carbon footprints.
- With stringent regulations and scrutiny from end customers and consumers, the shippers and consignees would prefer ports committed to a decarbonisation goal. Meeting such demands gives ports access to the global green market.
- Decarbonisation creates new job opportunities and thus accelerates economic growth. For example, the investment in wind and solar energy (renewables), research and development, and other innovative technologies revolutionize sustainable and green industries in these countries.
- Decarbonisation efforts improve air quality and public health in surrounding communities (socioenvironmental benefits). In other words, technologies that reduce $CO_2$, most of the time reduce other air pollutants as by-catch (e.g., NOx, SOx, PM, VOC).
- Implementing decarbonisation attracts international support, which can mitigate the issues (barriers) explained earlier. The global initiatives introduced by the IMO, the EU, and the C40 cities can provide the required technical and financial support.
- Developing countries can benefit from the international incentives, e.g., revenues of IMO MBMs, EU ETS, subsidies, and grants, among others, to support decarbonisation efforts and mitigate the financial costs. Thus, this engages them in the global realm of new businesses.
- Many ports, including developing countries' ports, can play a part in the value chain of the production and distribution of zero and near-zero $CO_2$ emission technologies, renewable fuels and energy sources for international shipping and ports decarbonisation. While this might be profitable, it also mitigates many barriers, such as not being distanced by the new MBMs (particularly least developed countries and island states), and lowering cost of transport by having frequent ship calls.
- Increase the security of energy supply. Ports that take part in energy transition (decarbonisation) can overcome insecurity in energy supplies posed by the current energy crisis in Europe, due to the Ukraine-Russia conflict, or the recent pandemic. As such, seaports need to have sustainable energy goals in their strategic plans.

## 7. Discussion and Conclusions

This study has reviewed the literature and built a framework that comprises the port decarbonisation concept and pathways, definitions, barriers and their solutions, including arising opportunities. The barriers were identified, categorised, and discussed based on previous frameworks that presented various categories which emerged from neoclassical economic theory perspective, such as the agency theory, and the contract theory. Therefore, this study is in line with various research that has identified and categorised the barriers, e.g., Refs. [58–60,64–74]. While these studies have categorised the barriers, they have not proposed solutions. Hence, this is a gap that this study has filled. Additionally, compared to other studies that tackled port decarbonisation, e.g., Refs. [28,34–38], this study aggregated

all the technologies and measures (concept and pathways), while at the same time proposed new (never addressed topic) definitions for port and maritime transport decarbonisation. Of consideration, while it is argued that port decarbonisation is a laborious issue, and saddled with challenges and obstacles (monetary, managerial, technical, social, etc.), this study pointed out how such challenging paradigms can be turned into opportunities that put the port on a better and more competitive track. Even for developing countries ports, there exist several opportunities that may better their environmental stance and improve their market position.

On this basis, the significance of this study lies in its manifold managerial and academic implications. The port managers, practitioners, and policy and decision makers benefit from the study as it highlights the reasons why ports may fail to uptake the decarbonisation measures and solutions to mitigate such challenges. As such, results support ports with tools to take reliable decisions and advance their decarbonisation performance. The solutions are also eye-openers for risk averse ports because the identified opportunities motivate all ports to prioritise decarbonisation as it is a very important global issue. Decarbonisation is the responsibility of all industries, sectors, governments, organisations and individuals because the climate change impact will harshly affect everyone everywhere, yet the price paid now for decarbonisation is far less than the future price. Of consideration, this study also contributes to the implementation of existing provisions, instruments, and regulations related to climate change such as the UNFCCC and its related legal instruments, including the Paris Agreement, and the United Nations 2030 Agenda for Sustainable Development. The whole research effort contributes to port sustainability performance (socioenvironmental perspectives). Academically, while the study has filled academic gaps, it has enriched the discussion of port decarbonisation. The barriers and solutions can be used as a checklist in case studies investigations to discover context-specific barriers. The barriers, solutions, and the definitions, which are expected to be highly used over the next few years, can be used by research for further empirical investigations and validation. The results can also be applied broadly, i.e., in shipping, inland waterways, and land transport decarbonisation. It is worth noting that this study is limited by its conceptual stance and is only based on literature review that is not systematic. While this is so, systematic review would not have worked as there are no studies on port decarbonisation and specifically on its barriers. Very few studies exist, and they were all included in the analysis. Notably, future research should include socio-economic dimension of decarbonisation as it is necessary to provide further options on viable ways of imposing a global $CO_2$ regulations and market measures. An exergoenvironmental analysis can also support in assessing the environmental responsibility of each port. Furthermore, how industries compete for decarbonising fuels and priority options should be future research areas.

Finally, based on this study result and analysis, the following policy recommendations are suggested for ports:

- Countries and ports need to develop decarbonization strategies that have proper policy packages and introduce grants, subsidies, investment, performance standards and mandates, communication and education campaigns, and carbon tax if needed.
- Goals and objectives of the decarbonisation strategy must be Specific, Measurable, Achievable, Realistic and Timebound (SMART) to facilitate implementation.
- Ports in different countries need to establish consistent and uniform environmental policies (standardisation).
- Ports need to conduct emission inventory. This is the first step and the cornerstone to decarbonisation.
- Ports need to solidify and prioritise education on technological and economic sides as well as boosting awareness for climate issues.
- Ports need to have mixes (combinations) of policies (solutions), e.g., regulations and disincentives (harsh), and incentives and capacity building (sweet but not sufficient).
- Decarbonisation comes with a price, Ports and supply chains should reduce this price to end customers and consumers as this issue may trigger political instability in

countries, leading to the rise of nationalist and populistic parties who don't prioritise decarbonisation, and thus decelerating implementation.

**Author Contributions:** Conceptualization, A.S.A.; methodology, A.S.A.; validation, A.S.A., D.D., F.B. and A.I.Ö.; writing—original draft preparation, A.S.A.; writing—review and editing, A.S.A.; visualization, A.S.A.; supervision, D.D., F.B. and A.I.Ö. All authors have read and agreed to the published version of the manuscript.

**Funding:** This research received no external funding.

**Institutional Review Board Statement:** Not applicable.

**Informed Consent Statement:** Not applicable.

**Data Availability Statement:** Not applicable.

**Acknowledgments:** The authors are thankful for the editor's support and reviewers' constructive comments.

**Conflicts of Interest:** The authors declare no conflict of interest.

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
