# Peer review of "Consolidating Port Decarbonisation Implementation: Concept, Pathways, Barriers, Solutions, and Opportunities"

_sustainability, doi:10.3390/su151914185_

Round 1

Reviewer 1 Report

The paper is well structured and complete. The literature review (which is the main objective) is rich and the discussion also points to interesting factors and needs. 

I have only two comments:

- The first is related to the form of the paper, because I know that sometimes the editor suggests to divide the discussion and the conclusions into two different sections, and you can take this into account!

- The second is related to the decision to write a review on this subject; I think that in order to promote a good use of this kind of work on existing sources and materials, it could be appropriate to anticipate at the beginning (introduction or first section) what are the main limitations of such a work (not the “your” but of the “genre”). I know that it should be taken for granted that it is a review, but I think that it could be useful for the authors to reflect on this.

Author Response

Thank you for this constructive comment. The responses are highlighted in green in the manuscript (R1) and also stated in the quotation below. We have added the following to the introduction.

“Ports play a crucial role in cities’ and urban areas' economic, environmental, and social sustainability, with the impact they have extending well beyond the waterfront. They are vital nodes in the urban logistics network, enabling the movement of products and commodities into and out of towns and cities. Ports considerably boost urban economies by creating employment opportunities, bolstering trade-related industries, and increasing local tax revenues. In addition, ports play a vital role in freight transportation, serving as crucial transfer and consolidation nodes that optimize supply chains and reduce transportation costs for urban businesses. The strategic location and role of ports in integrating global supply chains makes them indispensable elements of urban logistics, ensuring that cities remain economically robust and well-connected to the international markets.”

Reviewer 2 Report

This is a well executed review of the literature on port decarbonization, which also provides new policy recommendations.

At the beginning of Section 2, the ahuthors claim that only high-quality studies were included in their analysis and low quality and repetitive studies were not. What criterion was used to distinguish between high and low quality studies? Please be specific.

In my view, this is a long paper with a very large set of references, which would benefit greatly from being shorter and more concise.  

The paper also needs substantial work in terms of the use of the English language. There hundereds of typos in every section. Let me give you an example. In first two lines of pape 25, lines 906 and 907, I counted six typos. The words "significance" "managerial" "academic" "practitioners" "decision" and "from" were all misspelled. For the paper to be publishable, it needs to be substantially improved in the way it is written.

I have commented above that the paper needs to significantly improve the use of the English language and fix the numerous typos.

Author Response

Thank you for these constructive comments. The responses are highlighted in green in the manuscript (R1) and also stated in the quotation below.

  1. At the beginning of Section 2, the authors claim that only high-quality studies were included in their analysis and low-quality and repetitive studies were not. What criterion was used to distinguish between high and low-quality studies? Please be specific.

We have added this to the material and methods (highlighted in green)

“This yielded 241 studies. Duplicate studies were removed (99). The filtering process included two stages. Stage one focused on exclusion of irrelevant studies based on title and abstract reading, thus, those that were generic in nature (33) and didn’t meet the review question (23) were removed. The second stage analysed the paper by full reading, hence, only high-quality studies were included particularly those that met the objective of this study and had scientific rigor. Additionally, repetitive and low-quality studies were not considered. As a result, 86 studies were included.”

  1. In my view, this is a long paper with a very large set of references, which would benefit greatly from being shorter and more concise.  

Thank you very much for this comment. We agree that the paper is long. However, if the barriers and solutions are separated, it will not serve the purpose of the study. Additionally, we ain to have one stop shop for all the barriers and solutions to be eye opening for managers and researchers. We hope that the manuscript can be accepted as it is.

  1. The paper also needs substantial work in terms of the use of the English language. There hundereds of typos in every section. Let me give you an example. In first two lines of pape 25, lines 906 and 907, I counted six typos. The words "significance" "managerial" "academic" "practitioners" "decision" and "from" were all misspelled. For the paper to be publishable, it needs to be substantially improved in the way it is written.

We have conducted a thorough English proof reading and editing by English speaker.

Reviewer 3 Report

The authors have done exhaustive review on the port decarbonization, offered valuable implications for port managers, policy makers, practitioners, and other pertinent authorities.

1. Words in Figure 1 are not readable. The boxhead of Table 1 is missing.

2. Some subscripts are mislabeled. Page 5, “the amount of CO2 that would have the same global warming ability,”

3. There is no data to support the methods or techniques mentioned in the review. A comparison table is recommended to exhibit the advantages and disadvantages of various decarbonization strategies.

Minor editing.

Author Response

Thank you for these constructive comments. The responses are highlighted in green in the manuscript (R1) and also stated in the quotation below.

1.a. Words in Figure 1 are not readable.

We have improved the figure

1.b. The boxhead of Table 1 is missing.

We have added the table caption (Table 1 Port decarbonisation technical and operational measures )

2. Some subscripts are mislabeled. Page 5, “the amount of CO2 that would have the same global warming ability,”

All the subscripts were corrected           

3.a. There is no data to support the methods or techniques mentioned in the review.

We have added this to the material and methods (highlighted in green)

“This yielded 241 studies. Duplicate studies were removed (99). The filtering process in-cluded two stages. Stage one focused on exclusion of irrelevant studies based on title and abstract reading, thus, those that were generic in nature (33) and didn’t meet the review question (23) were removed. The second stage analysed the paper by full read-ing, hence, only high-quality studies were included particularly those that met the ob-jective of this study and had scientific rigor. Additionally, repetitive and low-quality studies were not considered. As a result, 86 studies were included.”

3.b. A comparison table is recommended to exhibit the advantages and disadvantages of various decarbonization strategies.

We have added one paragraph about the advantages and disadvantages of the decarbonization measures and also referred the readers for studies that focused on the measures per se. The barriers included many issues (disadvantages of technologies too), such as the technical barriers. The opportunities also included many advantages of the decarbonization technologies. (highlighted in green below Table 1)

“It should be noted that decarbonization measures are not a silver bullet to achieve zero emission ports. However, they can assist in the transition, through combinations of measures, to achieve the maximum abatement potential. In addition, the measures have challenges (technical issues, costs, and varying abatement potential, space requirement (high density of alternative fuels), high upfront cost, environmental benefits (the case of electrification based on fossil fuels grids), life cycle issues, methane and CO2 slip, the long lead time for infra-structure upgrading and renovations, etc.), see explanation of the barriers, in detail, in section 4. Consequently, ports should carefully evaluate the measures and consider advantages and disadvantages through feasibility and cost benefit ratio studies.”